# Accumulation of Lymphoid Progenitors with Defective B Cell Differentiation and of Putative Natural Killer Progenitors in Aging Human Bone Marrow

**DOI:** 10.3390/ijms262110467

**Published:** 2025-10-28

**Authors:** Laura Poisa-Beiro, Jonathan J. M. Landry, Aleksandr Cherdintsev, Michael Kardorff, Volker Eckstein, Laura Villacorta, Judith Zaugg, Anne-Claude Gavin, Vladimir Benes, Simon Raffel, Anthony D. Ho

**Affiliations:** 1Department of Medicine V, Heidelberg University, 69120 Heidelberg, Germany; laura.poisa.beiro@gmail.com (L.P.-B.); michael.kardorff@med.uni-heidelberg.de (M.K.); v.eckstein@t-online.de (V.E.); simon.raffel@med.uni-heidelberg.de (S.R.); 2Molecular Medicine Partnership Unit, EMBL & Heidelberg University, 69117 Heidelberg, Germany; judith.zaugg@unibas.ch; 3Genomics Core Facility, European Molecular Biology Laboratories (EMBL), 69117 Heidelberg, Germanylaura.villacorta@embl.de (L.V.); benes@embl.de (V.B.); 4Molecular Systems Biology, European Molecular Biology Laboratories (EMBL), 69117 Heidelberg, Germany; 5Department of Cell Physiology and Metabolism, University of Geneva, CH-1211 Geneva, Switzerland; anne-claude.gavin@unige.ch; 6Diabetes Center, Faculty of Medicine, University of Geneva, CH-1211 Geneva, Switzerland

**Keywords:** aging, senescence, human bone marrow, hematopoietic stem and progenitor cells, senolysis, immunosenescence, lymphoid compartment, natural killer progenitors, memory-like NK cells

## Abstract

In animal models, elimination of the senescent cells in the hematopoietic stem cells (HSCs) compartment leads to the rejuvenation of hematopoiesis. Whether this treatment principle can be applied to the human system remains controversial. The identification of senescent cells in human bone marrow poses another major challenge. To address these questions, we have studied hematopoietic stem and progenitor cells (HSPCs, CD34^+^) from the bone marrow of 15 healthy human subjects (age range: 19–74 years). Single-cell RNA sequencing, functional transcriptome analysis, and development trajectory studies were performed. In a previous report, we demonstrated the accumulation of a senescent population in the aging HSC compartment. The present study focuses on the differences with age downstream in the lymphoid trajectory. While a reduction in B progenitors in the early lymphoid compartment can be confirmed, the accumulation of a lymphoid cluster downstream upon aging is novel and remarkable. This cluster comprises cells with a significant deficiency in B differentiation markers, as well as 9.4% cells with transcriptome signatures of memory-like natural killer (NK) progenitors. Applying our analysis algorithm to other human bone marrow datasets from the literature, we are able to validate the presence of this unique cluster in aged lymphoid progenitors. The accumulation of a population comprising cells defective in B differentiation potential, as well as cells with transcriptome features of memory-like NK progenitors represents a novel hallmark for senescence in the late development trajectory of human lymphoid compartment.

## 1. Introduction

Aging has a grave impact on the composition and function of the immune system [1]. In the bone marrow, B lineage progenitors, including common lymphoid progenitors (CLP), pre-pro-B, and pro-B cells, undergo decline [1,2]. In contrast, myeloid progenitors increase in number and show only slight reduction in their proliferative potential [3,4]. The lineage skewing of hematopoietic stem and progenitor cells (HSPCs) towards myeloid development with age can be found in animal models as well as in human studies [3,4,5,6]. Comparative proteome and single-cell transcriptome studies between aged and young human HSPCs from our group have previously validated the lineage skewing in human HSPCs [7].

Several reports have demonstrated that aging of the hematopoietic system is caused by the accumulation of a senescent population [8,9]. Cellular senescence is associated with DNA damage, irreversible cell-cycle arrest, and a decline in biologic function in individual cells [8,9,10,11]. Within the lifespan of a cell, injuries, such as genomic instability and telomere attrition, accumulate [12,13,14,15]. Senescence plays an essential role for tissue homeostasis during normal embryogenesis [16] and prevents the proliferation of cells with DNA damage that might evolve into tumor cells within the aging process. Factors that induce cellular senescence include telomere attrition, DNA instability and mitochondrial dysfunction. The subsequent cell-cycle arrest prevents and suppresses tumor cell development [17,18]. In mice, telomere attrition and DNA damage induce one of two major tumor suppressive pathways, P53–P21 and p16Ink4a-pRb [8,9]. The activated Ink4a/Arf locus leads to sustained p53 activation and increased expressions of Cdkn2a and Cdkn2b [9,12]. Other alterations include reduced DNA damage response, chromatin remodeling, alterations in DNA methylation, elevated inflammation and stress responses and metabolic changes [19,20,21,22,23,24,25,26], in addition to the aforementioned lineage bias towards myelopoiesis [5,7]. Clearance of Cdkn2a and p16Ink4a positive cells has been shown to increase healthy lifespan and to improve aging disorders in mice [9,12,27].

The identification of senescent cells in human bone marrow poses, however, a major challenge. Despite several reports on successful therapy strategies targeting senescent markers in animal models, we do not have a clear concept on what the appropriate markers for senescence in humans are. Numerous studies have reported senescence-associated inflammatory phenotypes (SASPs) as markers for identifying senescent cells [28]. However, SASPs may represent more downstream markers [29]. Cellular senescence at different stages of development is probably associated with different underlying mechanisms. For mature B cells in circulating blood, senescence is characterized by a significant shift in B cell subsets, by a decrease in naïve B cells and an increase in memory B cells as well as age-associated B cells (ABCs) [1,2,28]. Cellular senescence in the lymphoid compartment of human bone marrow is, however, far less well studied.

In a previous report, we have demonstrated the accumulation of a senescent population in human HSC compartment with a strikingly similar senescence signature as in aged LT-HSCs from mice [29]. There was also a consistently significant increase in expression of CDKN1A in this senescent population, while the expression levels of CDKN2A and CDKN2B have been found to be very low or non-detectable at all ages, in contrast to the elevated expressions of Cdkn2a, Cdkn2b, and the activation of Ink4a/Arf and p53 in the aged murine HSCs.

The overarching goal of this study is to identify specific cellular and molecular markers that could identify senescent cells in the lymphoid trajectory of human hematopoiesis. We have provided evidence that, in analogy to the findings in mouse model, a small subset of senescent cells may be responsible for the aging process in the human hematopoietic stem cells (HSCs) compartment [27,29]. In this study, we focus on the lymphoid compartment in human bone marrow and have demonstrated that the accumulation of a distinct cluster of lymphoid progenitors further downstream represents a unique feature of aging. The molecular mechanisms of cellular senescence therefore vary with different stages of hematopoietic development.

## 2. Results

### 2.1. Single Cell Transcriptome Studies and the Developmental Trajectory of Human HSPC

We performed single-cell RNA sequencing studies on HSPCs (CD34+ cells) derived from 15 healthy human subjects (old subjects: *n* = 8; age range: 62–74 years; young subjects: *n* = 7; age range: 19–33 years). After quality control, normalization, principal components analysis and data projections using uniform manifold approximation and projection (UMAP), there were 24,567 genes identified [30].

We then applied the “Monocle 3” package within R to visualize the hierarchical continuum of developmental states [31,32,33,34]. Using markers as reported in the literature for classifying the developmental stages of HSPCs from primitive HSC to various committed progenitors, we annotated the cell clusters accordingly [35,36,37,38]. The results of this analysis for all of the 3784 HSPCs are summarized in Figure 1.

The present study focuses on mechanisms of aging during lymphoid differentiation. In Figure 2A, the trajectory of the lymphoid clusters, starting from common lymphoid progenitors (CLPs) is shown, in Figure 2B, that of the old, and in Figure 2C, that of the young subjects. The precise composition of the lymphoid clusters in the two age groups are summarized in Table 1. There are significantly more CLP, Ly1, Ly2 and Ly3 cells in the young than in the old subjects (*p* = 1.49 × 10^−5^, Chi-square test; see Table 1). In contrast, there is a significant accumulation of Ly4 in the old subjects. While 19.5% of lymphoid progenitors are found in the Ly4 cluster in the young subjects, this subset represents 49.9% of lymphoid progenitors in the old subjects (*p* = 1.95 × 10^−27^; Chi-square test). Cell cycle analysis showed that almost all of the cells in Ly4 are in G_o_ and in G_1_ phases (Figure 2D,E).

The developmental “pseudotimes” of the lymphoid clusters from the old are illustrated in Figure 3A, and those from the young subjects in Figure 3B. Figure 3C summarizes the side-by-side comparisons in pseudotimes between the two age groups in each of the clusters. Kolmogorov–Smirnov test shows that the delay in developmental time for Ly4 in aged human subjects is highly significant (*p* = 1.89 × 10^−8^) [39]. There is, therefore, a remarkable quantitative accumulation of Ly4, with a significant delay in the developmental trajectory.

### 2.2. Developmental Trajectory of Lymphocytic Clusters upon Aging

We then focused on the differences between the two age groups within each of the cell clusters, corresponding to various stages of B lymphocyte differentiation. Based on the transcriptomic profiles of B lymphocyte differentiation stages reported by Haddad R et al., Lee et al., Stewart et al., and Morgan & Tergaonkar, we have developed a list of 90 genes named “B cell development” that play a role in defining B cell developmental stages in humans [40,41,42,43]. The expression levels of individual genes in this gene set were mapped in the respective clusters along the development trajectory and depicted in detail in Appendix A. Figure 4A illustrates the expression profiles of 25 selected genes with conspicuous differences from this list (*y*-axis) in the respective clusters (*x*-axis). Based on the transcriptome profiles as depicted in Figure 4A and in Appendix A, we were able to re-classify the lymphoid clusters according to the respective transcriptome profiles. The corresponding differentiation stages are as follows: 1. Common lymphoid progenitors (CLPs); 2. Ly1 = Pre-Progenitor B cells (PrePro-Bs); 3. Ly2 = Progenitor B cells (Pro-Bs); 4. Ly3 = Precursor B1 cells (Pre-B1s), and 5. Ly4 = Precursor B2 cells (Pre-B2s) (See Table 1).

### 2.3. Differences Between Old and Young in Each of the Clusters in the Lymphoid Compartment

We then proceeded to compare the differences in expression of the gene set “B cell development” between the two age groups for each of the five lymphoid clusters. In Figure 4B, the results of these comparisons are summarized. The differences between the two age groups are most prominent in the Ly4 (Pre-B2) cluster (Figure 4B). A decrease in the following genes was found in the old human subjects: TCF3, CD24, CD22, LCN6, IFITM2, CD19, CD72, CD47, CIB1, SPN, and FLT3. The magnitude and the significance of the differences between the two age groups in these B markers are illustrated in the volcano plot Figure 4C, providing evidence for the defective B cell differentiation in the aged Ly4 (Pre-B2) population.

### 2.4. GSEA of the Lymphoid Clusters

Using gene set enrichment analysis (GSEA), we compared the statistical differences in gene expression profiles between the two age groups within each of the differentiation stages [44]. We searched for significant differences in the gene sets described by Hallmarks, Reactomes, KEGG, and Wikipathways [45,46,47,48]. The most prominent results are summarized in Figure 5. In the upper part, comparisons of expression levels from the old subjects versus young subjects as reference are shown. Significant elevations are found in the biological processes “DNA damage telomere stress induced senescence”, “Inhibition of DNA recombination at telomere”, and “DNA methylation”. In the lower part, the green shaded area, transcriptome expression levels from the young were compared with those from the old subjects as reference. When the lymphoid cells from all subjects were considered (the first column in Figure 5), the processes “B cell development”, “DNA repair” and “Mitotic spindle” were significantly elevated in the young, indicating a decline in B cell maturation, in repair of DNA damage, and in cell divisions in old human subjects.

We then analyzed the differences in each of the maturation stages. Comparisons between old and young subjects were performed in (a) CLP, (b) Ly1 (PrePro-B), (c) Ly2 (Pro-B), (d) Ly3 (Pre-B1), and (e) Ly4 (Pre-B2). Pathways that are significantly increased in the CLP compartment in old subjects include the following: “DNA damage telomere-stress induced senescence”, “Inhibition of DNA recombination at telomere”, and “DNA methylation”, demonstrating that some of the typical senescence signatures are found in the early lymphoid progenitors. Using the gene set “B cell development”, GSEA showed a significant reduction (enrichment score = −0.41; *p* < 0.001) in Ly4 (Pre-B2). Thus, the maturation defect in B development is significantly downstream in the development trajectory (Figure 5). The pathways DNA damage repair and mitotic spindle formation are also significantly impaired in the aged Ly4 cluster.

### 2.5. Accumulation of Memory-like NK Progenitors in Ly4 Compartment

Transcriptome studies of murine NK cells have indicated that they derive from the lymphoid progenitors in the bone marrow [49]. Single-cell RNA sequencing studies of NK cells isolated from peripheral blood and from bone marrow have recently revealed the heterogeneity of NK cells and NK progenitors (NKP) [50,51,52,53,54]. Applying the transcriptome signature for NKP as delineated by these authors and based on the respective expression levels, the individual lymphoid cells were scored and visualized in UMAP. We have identified 172 NKPs (13.4%) among all of the 1281 lymphoid progenitors examined, i.e., *n* = 87 (12.5%) of 695 lymphoid progenitors from old subjects versus *n* = 85 (14.5%) of 586 lymphoid progenitors from young subjects. Figure 6A depicts the detection of NKP in the lymphoid clusters, and Figure 6B the distribution of NKP in old versus young subjects. The most remarkable finding is a significantly higher proportion of NKPs in the aged Ly4 cluster (*p* < 0.001, Chi-squared test) (see Table 2).

Recent transcriptome studies have revealed the enormous heterogeneity of NK cells and NK cell progenitors in human bone marrow [52,53]. To determine whether there is a shift in the composition of NKP with age, we have applied the parameters reported by Yang et al. to categorize the cells accordingly [53]. As the numbers of NKP in the CLP clusters and in Ly3 clusters were too low for meaningful comparisons, we did not include the CLP cluster into the analysis and grouped Ly3 and Ly4 together for the differential expression analysis between the two age groups. The results are depicted in Appendix A. A significant increase in the expressions of B3GAT (CD57), CD52, CD69, and PMAIP1 in aged Ly3 + Ly4 cluster was found. No alterations of significance were discovered when comparing the two age groups in the clusters Ly1 and Ly2.

In a second step, we have applied the gene set reported by Guo et al. to identify the presence of putative memory-like NK cells (NK2.1) among the NKP found in our lymphoid clusters [55]. These authors discovered the accumulation of a memory-like proinflammatory NK subset that is predominantly CD52^+^NKG2C^+^CD94^+^ (corresponding to high gene expressions of CD52, KLRC2 & KLRD1 in our study) using single-cell transcriptome studies [55]. In vitro functional studies have indicated that these cells displayed a type I interferon response state. Notably, the accumulation of this subset correlated with the severity of COVID-19 pathology in the respective human subjects. In Figure 6C, the NKPs that were positive for the NK2.1 transcriptome score from our study are depicted, and in Figure 6D the distribution of these putative memory-like NK precursors in the two age groups. In summary, an increase in NKP, especially the putative, memory-like NK precursors, was found in the aged Ly4 population (Figure 6E). The dot-plot graph shown in Figure 6E illustrates prominent differences in the gene expressions of selected genes involved in NK and NK2.1 development between the two age groups, confirming the findings by Guo et al.

### 2.6. Accumulation of Ly4 Analogous Population in the Lymphoid Progenitors in Other Human Bone Marrow Datasets

Applying our algorithm, we analyzed the development trajectories of other human bone marrow datasets available in the literature [56,57]. We focused on the comparison of the lymphoid clusters from the two age groups, especially on the expressions of B cell development genes. Figure 7A shows the development trajectory of the HSPCs from the adult and aged human subjects derived from the dataset published by Zhang et al. [57]. We were able to identify the lymphoid clusters Ly1, Ly3, Ly4 (within the green circle). These clusters are extracted and shown in Figure 7B. In Figure 7C the trajectory of the young and in Figure 7D that of the old subjects are illustrated. The accumulation of an Ly4-analogous cluster in aged lymphoid progenitors is again identified.

We performed GSEA comparing the expression profiles of genes involved in B cell development between the two age groups in CLP as well as in Ly4-analogous clusters. The decrease in expression of B cell differentiation markers in the Ly4 population in old subjects is statistically significant in both external datasets (Figure 7I).

## 3. Discussion

The most conspicuous alteration in the lymphoid compartment of bone marrow HSPCs upon aging is the accumulation of a cluster downstream in the B cell trajectory. While the present study has confirmed a significant reduction in B cell progenitors in the early phase of lymphoid development, the increase of a cell cluster downstream in the developmental trajectory is novel and unexpected. This cluster is mapped in the vicinity of pre-B cells. In comparison to the pre-B cells in young subjects, however, they show a significantly lower expression of genes that are characteristic for B cell maturation. Pseudotemporal ordering reveals that there is a significant delay in the developmental time (pseudotime). Cell-cycle analysis shows that they are quiescent. Transcriptome analysis indicates that 9.4% of these cells express the characteristics for NK cell progenitors (NKPs), and half of these NKPs feature memory-like NK2.1 precursors. This increase in a unique subset of senescent cells in the aged lymphoid trajectory can be validated in two other human bone marrow datasets from the literature [56,57].

The present knowledge on the aging of the lymphoid compartment in the bone marrow has been based mainly on data from murine models [58]. Development trajectories from primitive HSCs to mature blood cells, including B lymphocytes, involve multiple stages [59]. Single-cell transcriptome studies in mouse models have provided insights into the mechanisms of aging at various stages of development [2,5,19,35,58,60]. In murine models of aging, the number of HSCs increases despite functional decline. There is a pronounced lineage skewing of HSCs towards myeloid development, and the lymphoid-primed progenitors such as CLPs decrease markedly [5,19,61,62]. Correspondingly, the pro-B, pre-B cells in bone marrow, and the B lymphocytes in peripheral blood drop dramatically in mice [61,62]. Studies in human bone marrow have confirmed the lineage skewing towards myeloid development but the change is more gradual [4,6,7]. The loss of lymphoid-primed progenitors in human bone marrow, as well as the decline in mature B cells in peripheral blood are far less pronounced [2,62,63]. While there are a number of reports on the aging of lymphocytes in human circulating blood, alterations in the lymphoid compartment in human bone marrow is far less studied [63,64]. The accumulation of a unique population in the lymphoid trajectory in human bone marrow with age has not been described in mouse models. Similarly, the accumulation of a distinct senescent cell cluster in the HSC trajectory with age is much more conspicuous in human subjects than in mice, as already reported in our previous study [29]. This accumulation of senescent cells downstream in the lymphoid trajectory, similar to the senescent population early in the HSC trajectory, seems to represent a human-specific phenomenon.

The accumulation of NK-like progenitors in the lymphoid compartment of human bone marrow, albeit to a much smaller extent, was another remarkable finding. The latter confirms the recent report by Guo et al. on the accumulation of NK2 cells that phenotypically resemble memory-like NK cells in aged human subjects [55]. These authors discovered a memory-like, proinflammatory NK subset that is predominantly CD52^+^NKG2C^+^CD94^+^, corresponding to the high expressions of CD52, KLRC2 and KLRD1 in the present study. They suggested that chronic overactivation of this subset may be responsible for the inflammaging as a conspicuous hallmark for the aging human lymphoid compartment [55]. Applying their transcriptome profile for identifying NK2.1 cells, we discovered that about half of the NKPs in the Ly4 subset express this transcriptome signature. In addition, Guo et al. performed in vitro functional studies, indicating that these cells displayed a type I interferon response state. Above all, they demonstrated that the accumulation of this subset correlated with the severity of COVID-19 pathology [55]. Given the small number of putative memory-like NKPs in our samples, we were not able to perform confirmatory functional studies. Our results correlate remarkably with the transcriptome data of memory-like NK cells, as reported by Guo et al. Further in-depth studies on the developmental trajectory of NKPs from HSPCs to memory-like NK2.1 cells, both in human bone marrow as well as in circulating blood, may provide more insight into the complex mechanisms of aging at different stages and on how to exploit this knowledge to target senescent cells in the late lymphoid trajectory.

In the aged human CLP population, GSEA showed a significantly elevated activity in the pathways “DNA damage telomere-stress induced senescence”, “Inhibition of DNA recombination at telomere”, and “DNA methylation”, and a significantly suppressed “DNA repair” process. This indicates that senescence in the early lymphoid progenitors is again characterized by DNA damage, telomere attrition, and impairment of DNA repair. In subsequent stages of the B cell development trajectory, GSEA confirms that senescence is characterized by the accumulation of cells with deficient expression of B markers, reduced DNA repair capacity, and of cells with transcriptome characteristics of memory-NK-progenitors (NK2.1 cells). Hence the mechanisms behind senescence and the respective markers at various stages of development are, with the exception of cell-cycle arrest, quite different.

In the HSC compartment of the bone marrow of animal models, an accumulation of senescent cells has been identified as a causal factor for aging [64,65,66,67,68]. Telomere attrition and DNA damage have been shown to induce one of two major tumor suppressive pathways, P53–P21 and p16Ink4a-pRb [66,67,68]. In contrast to experimental evidence provided by transplant and lineage tracing experiments in animal models, functional proof of senescence in humans is mostly indirect. In a previous report, we have nevertheless demonstrated the accumulation of a senescent population in a human HSC compartment with a strikingly similar senescence signature, as in aged LT-HSCs from mice [29]. There was, in addition, a consistently significant increase in the expression of CDKN1A in this senescent population, while the expression levels of CDKN2A and CDKN2B have been found to be very low or non-detectable at all ages. This is in contrast to the elevated expressions of Cdkn2a, Cdkn2b, and the activation of Ink4a/Arf and p53 in aged murine LT-HSCs. Senescence in the human HSC compartment is characterized by the activation of CDKN1A and with elevated P53–P21 pathway activity and not through the p16Ink4a-pRb pathway.

In mouse models, the primary site of lymphoid and NK cell development is the bone marrow. Early and primitive HSCs give rise to lymphoid-primed multipotent progenitors (LMP) and CLP, from which innate lymphoid cells and NK cells are then generated [51,53,54,68]. In all of these animal studies, NK cells from the bone marrow or from peripheral blood were isolated as starting material. In contrast, we have examined the HSPCs (CD34+ cells) from human bone marrow. Our focus was on the age-associated differences in the lymphoid progenitors. Our data indicate that significant changes between the two age groups are found in the Ly3 + Ly4 compartment, both quantitatively as well as qualitatively. Expressions of CD57 (B3GAT1), PMAIP1, CD69, and CD52 are prominently elevated in the aged NKPs of the Ly3 + Ly4 clusters. CD57 (B3GAT1) is a marker for cells that have reached their replicative end-stage and have lost their proliferative capacity. Simultaneously these cells have a higher expression of pro-inflammatory cytokines and are associated with chronic inflammation. PMAIP1 is a pro-apoptotic protein and a key mediator of the BCL-2 family and reflects the cellular stress and damage that accumulate with age. CD69 is linked to migration and retention in lymphoid organs and CD52 enhances trans-endothelial migration and activation of lymphocytes. Increased expression of CD52 and CD69 suggests a compensatory mechanism to overcome age-related decline in immune function and an excessive immune response associated with autoimmune diseases. Taken together, the combination of significantly increased B3GAT1 (CD57), PMAIP1, CD52 and CD69 in the aged NKPs at the pre-B developmental stage is consistent with immunosenescence with loss of proliferative capacity and compromised immune response, identified by high CD57 and PMAIP1, and with inflammaging, identified by high CD57 expression and risk of autoimmune diseases, indicated by elevated CD52 and CD69 expressions.

Our previous study has demonstrated that aging in the primitive HSC compartment is associated with the accumulation of a senescent subcluster that is characterized by increased telomere attrition, activation of P53 pathway, cell cycle arrest, and a remarkable up-regulation of CDKN1A [29]. While SASP has been regarded as the most prominent hallmark for senescence and as targets for senotherapeutics by other authors [10,11], we have demonstrated that activation of this pathway was found only further downstream in more differentiated stages such as GMP. In analogy, we have provided evidence that cellular senescence at the level of CLP stage is characterized by DNA damage, telomere attrition and impairment of DNA repair, the accumulation of a subset with deficiency in B cell differentiation and of an NKP subset with transcriptome signature of immunosenescence. The mechanisms and consequently the “markers” for senescence can be very different at specific stages of differentiation. A precise understanding of the molecular and cellular mechanisms that lead to the accumulation of senescent cells at a specific level of development is therefore a prerequisite for targeting senescent cells.

## 4. Materials and Methods

### 4.1. Human Specimen

Bone marrow samples were harvested from healthy human subjects through puncture at the posterior iliac crest. The study has been approved by the Ethics Committee for Human Subjects, University of Heidelberg, and written informed consent was obtained from each individual (Approval code: S-480-2011, Date: 3 June 2019), [7,29]. The cohort of 15 human subjects was categorized according to age into two classes: old (≥59 years, *n* = 8, median = 67 years), and young (≤31 years, *n* = 7, median = 30 years). Detailed features are summarized in Appendix A. The preparation of human specimens and the method for the isolation of CD34+ cells are published in detail in [7,29].

### 4.2. Single-Cell RNA Sequencing of HSPCs

Sequencing libraries from the human HSPCs were generated based on the smart-seq2 protocol of Picelli et al. [69] and the tagmentation procedure of Hennig et al. [70]. The details are published in [7,29].

### 4.3. Data Processing

The single-cell data preprocessing was performed using the programming language R. Raw reads were aligned using STAR aligner [71]. The count matrix generated for individual genes across cells in each sample was then subjected to further processing using the Bioconductor package Seurat V4 [72]. The details are described in previous publications [7].

### 4.4. Toolkit for Analyzing Single-Cell Transcriptome Data

We apply the toolkit Monocle 3 package within the R environment for analyzing single-cell gene expression data [30,31,32,33,34]. Dimension reduction was based on uniform manifold approximation and projection (UMAP) [30], followed by clustering of cells. The workflow consists of organizing cells into trajectories, followed by statistical tests to identify genes that vary in expression over those trajectories [32,33,34]. The details are provided in previous publications [7,26,29]. To verify the trajectory and pseudotime inference, we have also applied the toolkit “slingshot” for analysis [73]. As the results confirmed the accumulation of a unique subset downstream in the lymphoid trajectory as using Monocle 3.

Gene sets used for annotation of cell clusters:

The gene sets used to classify and to define the lineage-specific signatures for the HSPCs are listed in previous publications [26,29].

The following transcriptomic markers were used to identify NK progenitors in CD34+ BM cells: CD34, CD7, PTPRC, IL7R, KIT, ID2, NFIL3, TOX, TBX21, IL7RA [50,51,53,54]. The presence of these markers was checked in individual cells and scored based on normed gene expression levels. A score of >0.4 was considered to be positive and the respective cell mapped in UMAP.

For the detection of putative memory-like NK precursors, the following transcriptome markers were used: IRF7, IRF9, POLR2A, EZH2, POU3F1, STAT1, HMGB2, KLF6, KLF10, KLRC2, KLRD1, CD52. A score of >0.4 was considered to be positive and the respective cell mapped in UMAP.

### 4.5. Cell Cycle Annotation

Each cell was scored for S phase and G2 phase using the marker gene list for the respective phase (Seurat package, extracted from Tirosh et al. [74]). Using these 2 scores, each cell was then assigned to G0, G1, S and G2-M according to Kowalczyk et al. [19].

### 4.6. Gene Set Enrichment Analysis (GSEA)

We applied the GSEA (version 4.3.2., Broad Institute, Inc., Massachusetts Institute of Technology) for interpreting gene expression data [44,45]. We searched for significant differences in the gene sets defined in Hallmarks, Reactomes, KEGG, and Wikipathways [45,46,47,48].

### 4.7. Generating the Gene Set “B Cell Development”

Based on the transcriptomic profiles of B lymphocyte differentiation stages reported by Haddad R et al. [40], Lee et al. [41], Stewart et al. [42], and Morgan & Tergaonkar [43], we have developed a list of 90 genes named “B cell development” that play a role in defining B cell developmental stages in humans.

### 4.8. Data Availability

Raw data for the single-cell RNA seq of human specimens have been deposited in the European Nucleotide Archive (ENA) database under accession ID PRJEB68076. The authors declare that all data supporting the findings of this study are available upon reasonable request.

## 5. Conclusions

Our previous data have provided evidence that aging in the early HSC compartment is caused by the clonal evolution of a population characterized by increased telomere attrition, cell-cycle arrest, and a remarkable up-regulation of CDKN1A [29]. Activated SASP and dysregulation of DNA methylation represent mechanisms that are found more downstream, at the level of early myeloid development [29]. In this study, we have demonstrated that, while telomere attrition and DNA damage may represent signatures of senescence in the CLP cluster in the early lymphoid development, an accumulation of a unique lymphoid cell cluster represents a prominent feature of aging in the lymphoid trajectory. This senescent population is characterized by lymphoid cells defective in B cell development, as well as by putative natural killer progenitors with transcriptome features of memory-like NK precursors. Applying our analysis algorithm, we were able to validate the accumulation of this subset in other human bone marrow datasets in the literature. The elimination of senescent cells at specific stages of development therefore requires a precise understanding of the underlying molecular and cellular mechanisms of aging at the respective level for the development of appropriate senolytic treatment strategies.

## Figures and Tables

**Figure 1 ijms-26-10467-f001:**
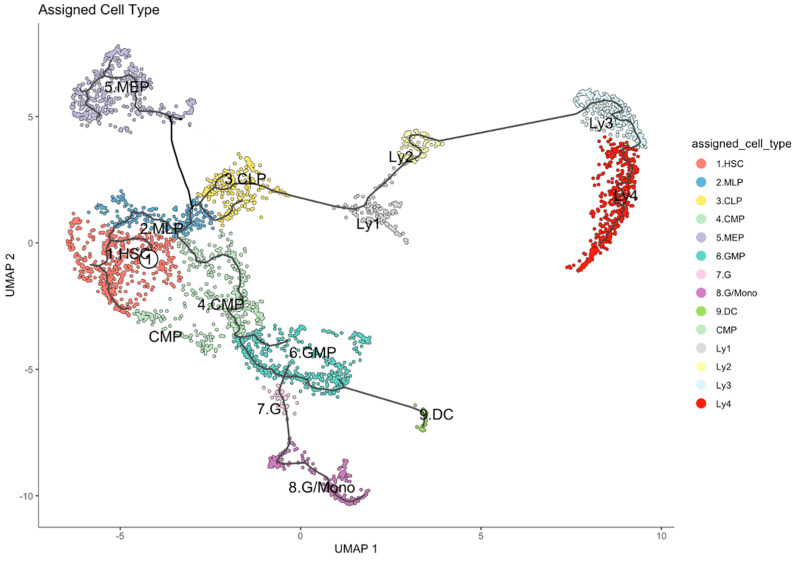
Developmental trajectory of the HSPCs (CD34+ cells) of 15 healthy human subjects. Starting with the root (white circle 1), the development from the primitive HSC compartment to myeloid–lymphoid progenitors (MLP), megakaryocte and erythrocyte progenitors (MEP), granulocyte–monocyte progenitors (GMP), and precursors for granulocytes (G), monocytes (M) and dendritic cells (DC), as well as to various precursors of B lymphocytes (Ly1 to Ly4). The black lines represent the structure of the graph.

**Figure 2 ijms-26-10467-f002:**
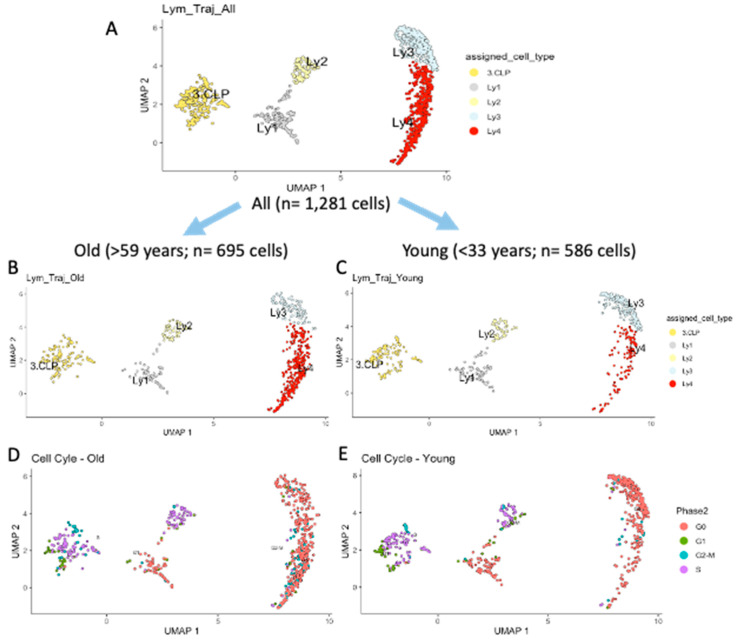
The development trajectory of the lymphoid compartment in human bone marrow. (**A**) shows the trajectory for all lymphoid progenitors, (**B**) illustrates the developmental stages in old subjects, (**C**) the developmental stages in young subjects, (**D**) the cell-cycle phase of the cells in old subjects, and (**E**) the cell-cycle phase of the cells in young subjects.

**Figure 3 ijms-26-10467-f003:**
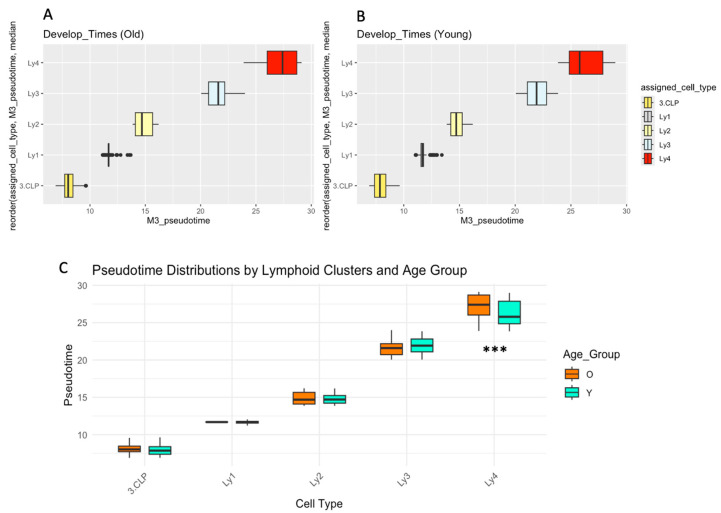
The hypothetical developmental duration, “pseudotimes”, of the lymphoid clusters. (**A**) The precise “pseudotimes” of each of the lymphoid clusters in the old subjects, (**B**) the pseudotimes of the lymphoid clusters in the young subjects, (**C**) the results of statistical analysis in the old compared with young human subjects. The Kolmogorov–Smirnov test showed a significant delay in the median “pseudotimes” of the “Ly4” cluster in old subjects (box plots with *** = *p*-values of <0.001).

**Figure 4 ijms-26-10467-f004:**
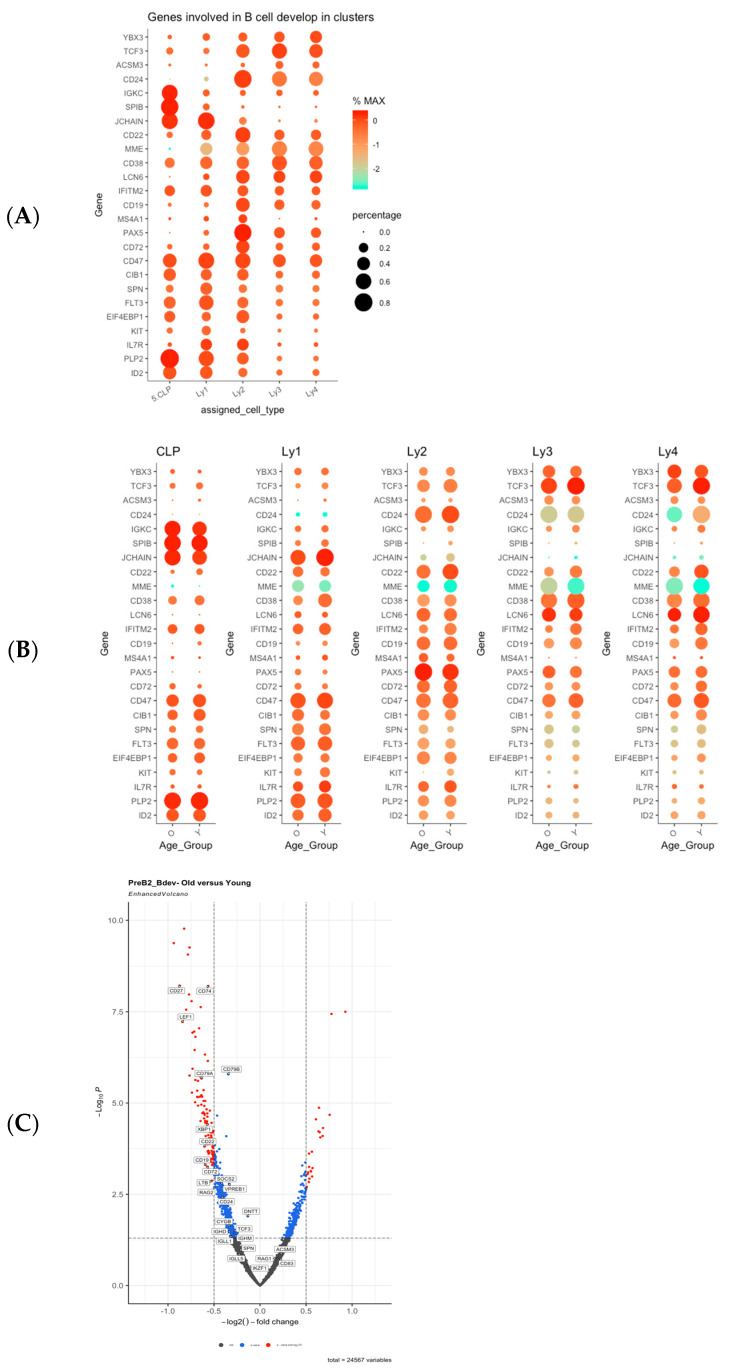
Transcriptome expression profiles of the lymphoid clusters: (**A**) Transcriptome expression profiles of selected genes from the list “B cell development” at differentiation stages along the B trajectory. (**B**) The age-dependent alterations of selected genes within the list “B cell development”. The differences in gene expression levels between the two age groups within each of the respective developmental stages from Ly1 to Ly4 are shown. Decreased expressions of genes such as TCF3, CD24, CD22, LCN6, IFITM2, CD19, CD72, CD47, CIB1, SPN, and FLT3 in the aged Ly4 clusters are more evident. (**C**) Volcano plot of the differential expressions between old and young in the Ly4 cluster, corresponding to PreB2 development stage. The results confirm that, in the Ly4 lymphoid cluster, the reduction in expression of B cell development genes such as CD27, CD74, LEF1, CD79A, CD79B, XBP1, CD22, CD19, CD72, LTB, and RAG2 in old human subjects becomes statistically significant and in higher magnitude.

**Figure 5 ijms-26-10467-f005:**
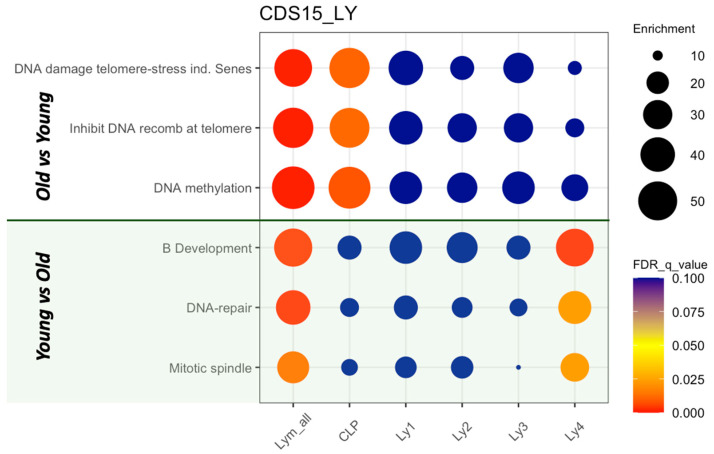
GSEA of expression profiles in pathways associated with aging—comparisons between old and young subjects in lymphoid clusters of human HSPCs. In the upper part, comparisons of expression levels from the old subjects versus young subjects as reference are shown. In the lower part, the green shaded area, transcriptome expression levels from the young are compared with those from the old subjects as denominator.

**Figure 6 ijms-26-10467-f006:**
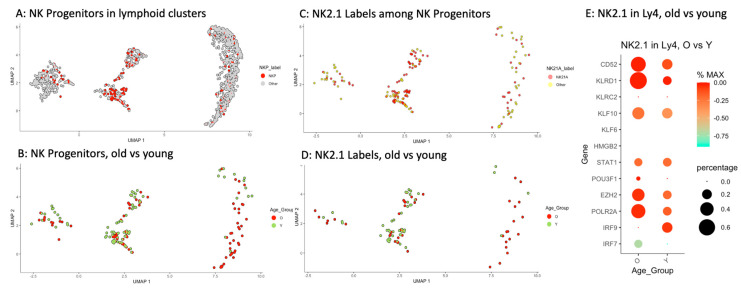
Detection of progenitors with transcriptome profiles of NKP in the lymphoid clusters applying the transcriptome signature for NKP as delineated by other authors, 13.4% of all of the 1281 lymphoid progenitors were scored positive for NK progenitors. (**A**) depicts the mapping of putative NKPs in the lymphoid clusters, and (**B**) the distribution of NKPs in old versus young subjects. The most remarkable finding is a significantly higher proportion of NKPs in the aged Ly4 cluster (*p* < 0.001, Chi-squared test). (**C**) The cells that were scored positive for NK2.1 signature among the NKPs. (**D**) The distribution of putative NK2.1 cells in old versus young subjects. (**E**) The relative expression levels of some selected individual genes characteristic for memory-like NK2.1 precursors.

**Figure 7 ijms-26-10467-f007:**
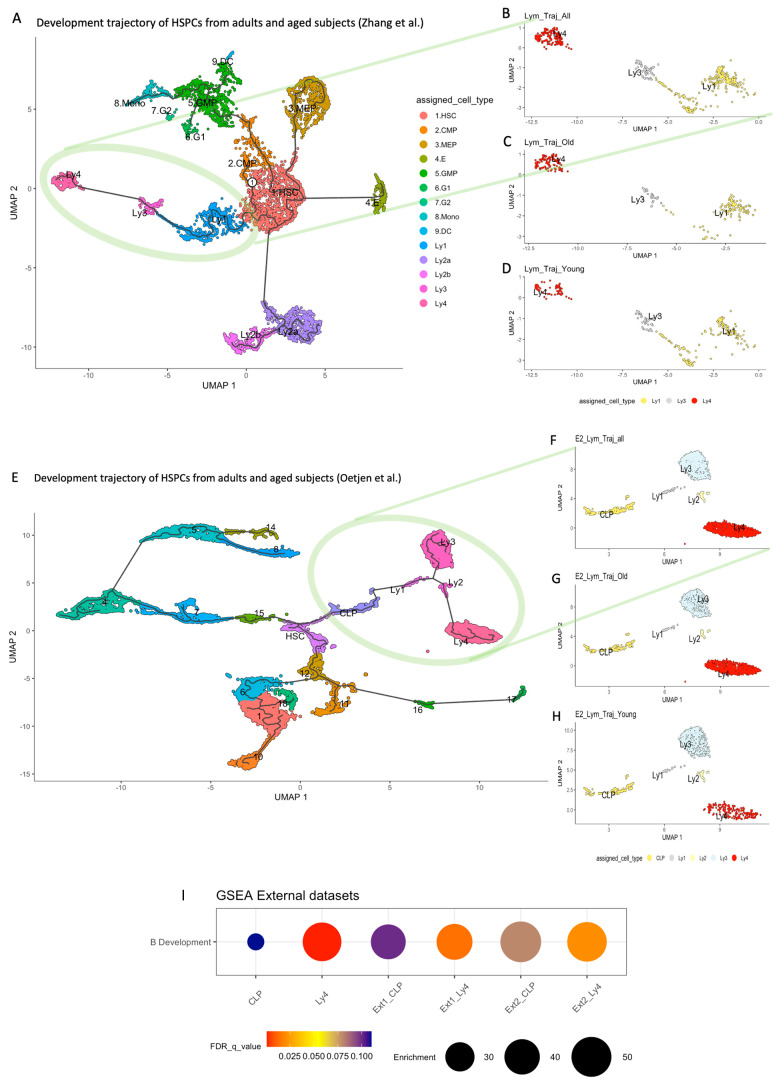
Validation of the Ly4 senescent cluster in other human bone marrow datasets. (**A**) The development trajectory of the HSPCs from the adult and aged human subjects, from the dataset published by Zhang et al. [57]. Using our algorithm, the lymphoid clusters Ly1, Ly3, Ly4 (within the green circle) could be identified. (**B**) The trajectory of the lymphoid clusters Ly1, Ly3 and Ly4. (**C**) The trajectory in the young and in (**D**) the old subjects. The accumulation of an Ly4-analogous cluster in aged lymphoid progenitors is identified. (**E**) The development trajectory from the young adults between the age range of 19 to 33 years, and from the old individuals with age beyond 59, extracted from the study of Oetjens et al. [56]. (**F**) The lymphoid clusters from all of the subjects. (**G**) The clusters from the old and (**H**) the young subjects. In both external datasets we are able to validate the accumulation of an Ly4-analogous population in the aged lymphoid progenitors. (**I**) The GSEA comparing the expression profiles of genes involved in B cell development. The reduction in expression of B cell differentiation markers in the aged Ly4 population is statistically significant.

**Table 1 ijms-26-10467-t001:** Composition of the lymphoid clusters in all 15 subjects, in the old (*n* = 8) and in the young (*n* = 7) human subjects. (Statistical analysis: *** = <0.001, n.s. = not significant, Chi-squared test).

Lymphoid Clusters	Re-Defined Lymphoid Clusters	All *n* = 1281 Cells %	Old *n* = 695 Cells %	Young *n* = 586 Cells %	*p* Value
**CLP**	**CLP**	20.2	18.1	22.7	n.s.
**Ly1**	**PreProB**	10.9	9.5	13.1	n.s.
**Ly2**	**ProB**	10.1	9.2	11.3	n.s.
**Ly3**	**PreB1**	22.8	13.8	33.4	***
**CLP+Ly1+** **Ly2+Ly3**	**Early Progenitors**	64.0	50.1	80.5	***
**Ly4**	**PreB2**	36.0	49.9	19.5	***

**Table 2 ijms-26-10467-t002:** Cells positive for the NK progenitor score defined by transcriptome expression in the lymphoid clusters. (Statistical analysis: * = <0.05, *** = <0.001, n.s. = not significant, Chi-squared test).

	Old			Young			*p* Values	
	Total n	NKP	NK2.1	Total n	NKP	NK2.1	NKP	NK2.1
**CLP**	126	12	6	133	14	4	n.s.	n.s.
**Ly1**	62	25	14	77	42	27	*	*
**Ly2**	64	12	10	66	16	11	n.s.	n.s.
**Ly3**	96	5	2	196	10	3	n.s.	n.s.
**Ly4**	347	33	15	114	3	1	***	***

## Data Availability

Raw data for the single-cell RNA seq of human specimens have been deposited in the European Nucleotide Archive (ENA) database under accession ID PRJEB68076. The authors declare that all data supporting the findings of this study are available upon reasonable request.

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
