# Peer review of "Accumulation of Lymphoid Progenitors with Defective B Cell Differentiation and of Putative Natural Killer Progenitors in Aging Human Bone Marrow"

_ijms, 2025, doi:10.3390/ijms262110467_

Round 1
Reviewer 1 Report
Comments and Suggestions for Authors
The manuscript submitted by Beiro et al. explores age-related alterations in the human hematopoietic stem and progenitor cell (HSPC) compartment using single-cell RNA-sequencing and trajectory analyses of CD34+ bone marrow cells. The authors confirm the expected decline in B progenitors with aging but further report a novel lymphoid cluster enriched in older subjects. This cluster is described as deficient in B-cell differentiation markers and partially composed of cells with transcriptomic signatures resembling memory-like NK progenitors.
While the identification of such a population is potentially significant, the study would benefit from further clarification regarding its functional relevance compared with murine studies, and how these findings advance the understanding of human hematopoietic senescence beyond descriptive observations.
Major concerns are listed below.
- Since the present study is focused exclusively on human samples, it would be valuable for the authors to more explicitly discuss on how human HSC/lymphoid aging differs from murine models. In particular, clarification of differences in lymphoid differentiation trajectories, subtype composition, and stage-specific markers between human and mouse systems would strengthen the translational significance of the findings. This comparison in discussion could help readers understand whether the novel lymphoid cluster identified here represents a human-specific phenomenon or has parallels in murine hematopoiesis.
- While the single-cell transcriptomic data provide compelling associations, the interpretation that this NKP-like population represents “immunosenescence” remains largely descriptive. Functional assays—such as differentiation capacity, cytokine secretion, or cytotoxic activity—would substantially strengthen the biological significance of these findings. In the absence of such validation, the conclusions should be moderated to emphasize correlation rather than definitive functional impairment.
3.The study relies on Monocle 3 for pseudotime and trajectory inference. While this is an established approach, trajectory analyses are inherently assumption-heavy and may yield method-dependent results. It would strengthen the robustness of the findings if the authors could demonstrate that the observed developmental delay and accumulation of the lymphoid cluster are consistent across alternative trajectory inference algorithms (e.g., Slingshot, PAGA, or other commonly used methods). If not feasible, the limitations of relying on a single method should be acknowledged.
- The section argues that senescence markers vary depending on the developmental stage. This is an important point, but the markers used (e.g., CD57, PMAIP1, CD52, CD69) can also reflect activation or stress responses rather than true senescence. Some caution in interpretation is warranted.
- The link between increased CD57/CD52/CD69 expression and inflammaging is intriguing, but the causal relationship remains speculative. It might be more precise to describe this as “consistent with” inflammaging rather than a defining hallmark.
Author Response
We thank Reviewer 1 for his constructive comments and suggestions for revisions. We appreciate his comment that the identification of such a population in the lymphoid trajectory is potentially significant.
Major concerns:
- Since the present study is focused exclusively on human samples, it would be valuable for the authors to more explicitly discuss on how human HSC/lymphoid aging differs from murine models. In particular, clarification of differences in lymphoid differentiation trajectories, subtype composition, and stage-specific markers between human and mouse systems would strengthen the translational significance of the findings. This comparison in discussion could help readers understand whether the novel lymphoid cluster identified here represents a human-specific phenomenon or has parallels in murine hematopoiesis.
This point is well taken and we have made revisions in the “Introduction” as well as in the “Discussion” to address this issue.
(Lines 318 to 335 in revised manuscript) - In murine models of aging, the number of HSCs increases despite functional decline. There is a pronounced lineage skewing of HSCs towards myeloid development, and the lymphoid-primed progenitors such as CLPs decrease markedly [5,19,61,62]. Correspondingly, the pro-B, pre-B cells in bone marrow, and the B lymphocytes in peripheral blood drop dramatically in mice [61,62]. Studies in human bone marrow have confirmed the lineage skewing towards myeloid development but the change is more gradual [4,6,7]. The loss of lymphoid-primed progenitors in human bone marrow, as well as the decline in mature B cells in peripheral blood are far less pronounced [2,62,63]. While there are a number of reports on aging of lymphocytes in human circulating blood, alterations in the lymphoid compartment in human bone marrow is far less studied [63,64]. The accumulation of a unique population in the lymphoid trajectory in human bone marrow with age has not been described in mouse models. Similarly, the accumulation of a distinct senescent cell cluster in the HSC trajectory with age is much more conspicuous in human subjects than in mice, as already reported in our previous study [29]*. This accumulation of senescent cells downstream in the lymphoid trajectory, similar to the senescent population early in the HSC trajectory, seems to represent a human-specific phenomenon. - While the single-cell transcriptomic data provide compelling associations, the interpretation that this NKP-like population represents “immunosenescence” remains largely descriptive. Functional assays—such as differentiation capacity, cytokine secretion, or cytotoxic activity—would substantially strengthen the biological significance of these findings. In the absence of such validation, the conclusions should be moderated to emphasize correlation rather than definitive functional impairment.
We acknowledge that due to the scarcity of these cells, in-depth analysis applying functional assays was not feasible at this juncture. We have, however, found the same accumulation of NKP-like population that represented “immunosenescence” as Guo et al. who have performed functional assays of these cells and have also notably found a significant correlation between the accumulation of this subset with the severity of COVID-19 pathology. This issue is addressed thoroughly in the “Discussion”.
(Lines 335 to 354 in revised manuscript) - The accumulation of NK-like progenitors in the lymphoid compartment of human bone marrow, albeit to a much smaller extent, was another remarkable finding. The latter confirms the recent report by Guo et al. on the accumulation of NK2 cells that phenotypically resemble memory-like NK cells in aged human subjects [55]. These authors discovered a memory-like, proinflammatory NK subset that are predominantly CD52+NKG2C+CD94+, corresponding to high expressions of CD52, KLRC2 & KLRD1 in the present study. They suggested that chronic overactivation of this subset may be responsible for the inflammaging as a conspicuous hallmark for the aging human lymphoid compartment[55]. Applying their transcriptome profile for identifying NK2.1 cells, we discovered that about half of the NKPs in the Ly4 subset express this transcriptome signature. In addition, Guo et al. performed in vitro functional studies, indicating that these cells displayed a type I interferon response state. Above all, they demonstrated that the accumulation of this subset correlated with the severity of COVID-19 pathology [55]. Given the small number of putative memory-like NKPs in our samples, we were not able to perform confirmatory functional studies. Our results correlate remarkably with the transcriptome data of memory-like NK cells reported by Guo et al. Further in-depth studies on the developmental trajectory of NKPs from HSPCs to memory-like NK2.1 cells, both in human bone marrow as well as in circulating blood may provide more insight into the complex mechanisms of aging at different stages and on how to exploit this knowledge to target senescent cells in the late lymphoid trajectory. - The study relies on Monocle 3 for pseudotime and trajectory inference. While this is an established approach, trajectory analyses are inherently assumption-heavy and may yield method-dependent results. It would strengthen the robustness of the findings if the authors could demonstrate that the observed developmental delay and accumulation of the lymphoid cluster are consistent across alternative trajectory inference algorithms (e.g., Slingshot, PAGA, or other commonly used methods). If not feasible, the limitations of relying on a single method should be acknowledged.
We have applied the toolkit „slingshot“ for trajectory and pseudotime inference and were able to confirm the accumulation of a unique senescent lymphoid subset in the aged human subjects, as well as the protracted pseudotime. However, the results are more evident using Monocle 3. Therefore, we did not present these results in the main manuscript. This is mentioned under “Methods”.
(Lines 443 to 446) To verify the trajectory and pseudotime inference, we have also applied the toolkit “slingshot” for analysis [73]. As the results confirmed the accumulation of a unique subset downstream in the lymphoid trajectory as using Monocle 3, these results are not shown in the manuscript. - The section argues that senescence markers vary depending on the developmental stage. This is an important point, but the markers used (e.g., CD57, PMAIP1, CD52, CD69) can also reflect activation or stress responses rather than true senescence. Some caution in interpretation is warranted.
We agree that the markers mentioned can also reflect activation or stress responses. Our line of thought was that, within the senescent lymphoid subset unique to aged healthy human subjects—particularly in the NKP-like cells—it is remarkable that these markers were expressed at significantly higher levels. This additional information supports our hypothesis of accumulation of lymphoid cells and NKP-like cells that correlates with clinical development of immunosenescence and inflammaging.
(Lines 385 to 389) - Our focus was on the age-associated differences in the lymphoid progenitors. Our data indicate that significant changes between the two age groups are found in the Ly3+Ly4 compartment, both quantitatively as well as qualitatively. Expressions of CD57 (B3GAT1), PMAIP1, CD69, and CD52 are prominently elevated in the aged NKPs of the Ly3+Ly4 clusters.
- The link between increased CD57/CD52/CD69 expression and inflammaging is intriguing, but the causal relationship remains speculative. It might be more precise to describe this as “consistent with” inflammaging rather than a defining hallmark.
(Lines 397 to 402) Taken together, the combination of significantly increased B3GAT1 (CD57), PMAIP1, CD52 and CD69 in the aged NKPs at the pre-B developmental stage is consistent with immunosenescence with loss of proliferative capacity and compromised immune response, identified by high CD57 and PMAIP1, and with inflammaging, identified by high CD57 expression, and risk of autoimmune diseases, indicated by elevated CD52 and CD69 expressions.
* Poisa-Beiro, L.; Landry, J.J.M.; Yan, B.; Kardorff, M.; Eckstein, V.; Villacorta, L.; Krammer, P.H.; Zaugg, J.; Gavin, A.C.; Benes, V.; et al. A Senescent Cluster in Aged Human Hematopoietic Stem Cell Compartment as Target for Senotherapy. Int J Mol Sci 2025, 26, doi:10.3390/ijms26020787
Reviewer 2 Report
Comments and Suggestions for Authors
The article “Accumulation of lymphoid progenitors with defective B cell differentiation and of putative natural killer progenitors in aging human bone marrow” indeed sounds very timely and significant. The authors rightly note that most data on hematopoietic aging have been obtained in mice. Their study partially fills this gap by using human samples, which is critically important for the clinical translation of potential therapies (such as senolytics). The use of single-cell RNA-seq represents the gold standard for studying heterogeneous populations such as HSPCs. The paper does more than merely report a reduction in B-cell numbers—it identifies the accumulation of a specific population at the late stages of lymphoid differentiation. This shifts the paradigm from “simple depletion” to “dysregulated differentiation and emergence of aberrant cells.” Despite the novelty and clear significance of the study, several questions arise:
-
Single-cell RNA-seq is a powerful tool, but it is well known that the transcriptome does not always fully reflect the proteome. It would be interesting to confirm the existence of this cluster using flow cytometry based on the identified marker genes. Such validation would strengthen the findings.
-
Is the lymphoid cluster that accumulates with aging truly dysfunctional? To make a definitive claim about impaired differentiation potential, functional assays in vitro or in vivo (e.g., transplantation into immunodeficient mice) are needed to directly demonstrate that these cells indeed fail to produce mature B cells.
-
Do cells in this newly identified lymphoid cluster display the same features of cellular aging as aging HSPCs (for instance, in the CLP population), or is this a fundamentally different type of “aging”? The study shows that cells in this cluster are quiescent and exhibit a delay in pseudotime. Do you consider this a genuine senescent state, or rather a reversible state of dormancy or differentiation block? Are there transcriptomic signatures in your data that would support one interpretation over the other?
-
Based on your findings, is it possible to propose specific markers that could enable selective elimination of this population of aging lymphoid progenitors while sparing healthy cells?
-
Can your results suggest why this cluster forms specifically at this stage of differentiation? Is it a consequence of intrinsic defects within the progenitors themselves, or the result of influences from the aging niche?
Author Response
We thank Reviewer 2 for his enthusiastic comments and very constructive critiques. This comment was deeply appreciated, “…the paper does more than merely report a reduction in B-cell numbers—it identifies the accumulation of a specific population at the late stages of lymphoid differentiation. This shifts the paradigm from simple depletion to dysregulated differentiation and emergence of aberrant cells.”
Questions:
- Single-cell RNA-seq is a powerful tool, but it is well known that the transcriptome does not always fully reflect the proteome. It would be interesting to confirm the existence of this cluster using flow cytometry based on the identified marker genes. Such validation would strengthen the findings.
This point is well received. Further in-depth analysis of this cluster with flow cytometry might have strengthened the manuscript. The challenge lies in that the significant differences in marker genes for the senescent population did not include surface antigens, (with the exception of the very rare NKP-like cells). We have therefore not been able to design a method to separate these cells for flow cytometry studies.
We have, however, previously performed a series of studies comparing proteomics and transcriptomic data from human HSPCs within the publication “Cell specific proteome analyses of human bone marrow reveal molecular features of age-dependent functional decline” *. We have found an outstanding correlation between proteome and transcriptome data in human HSPCs.
- Is the lymphoid cluster that accumulates with aging truly dysfunctional? To make a definitive claim about impaired differentiation potential, functional assays in vitro or in vivo (e.g., transplantation into immunodeficient mice) are needed to directly demonstrate that these cells indeed fail to produce mature B cells.
While we agree that functional assays would have provided additional evidence and validation, functional assays were not feasible at this juncture due to (1) the scarcity of and (2) the lack of surface markers for isolation of the senescent population. The transcriptome profile as well as the GSEA analyses all indicate that these cells correlate with deficiency in B differentiation. Furthermore, our transcriptome data on memory-like NK cells correlate remarkably with those reported by Guo et al., who performed functional studies on this population (NK-progenitors) derived from the peripheral blood. They also showed that this subset correlated with the severity of COVID-19 pathology. To address this issue, we have revised the “Discussion” part.
(Lines 335 to 350) - The accumulation of NK-like progenitors in the lymphoid compartment of human bone marrow, albeit to a much smaller extent, was another remarkable finding. The latter confirms the recent report by Guo et al. on the accumulation of NK2 cells that phenotypically resemble memory-like NK cells in aged human subjects [55]. These authors discovered a memory-like, proinflammatory NK subset that are predominantly CD52+NKG2C+CD94+, corresponding to high expressions of CD52, KLRC2 & KLRD1 in the present study. They suggested that chronic overactivation of this subset may be responsible for the inflammaging as a conspicuous hallmark for the aging human lymphoid compartment[55]. Applying their transcriptome profile for identifying NK2.1 cells, we discovered that about half of the NKPs in the Ly4 subset express this transcriptome signature. In addition, Guo et al. performed in vitro functional studies, indicating that these cells displayed a type I interferon response state. Above all, they demonstrated that the accumulation of this subset correlated with the severity of COVID-19 pathology [55]. Given the small number of putative memory-like NKPs in our samples, we were not able to perform confirmatory functional studies. Our results correlate remarkably with the transcriptome data of memory-like NK cells as reported by Guo et al.
- Do cells in this newly identified lymphoid cluster display the same features of cellular aging as aging HSPCs (for instance, in the CLP population), or is this a fundamentally different type of “aging”? The study shows that cells in this cluster are quiescent and exhibit a delay in pseudotime. Do you consider this a genuine senescent state, or rather a reversible state of dormancy or differentiation block? Are there transcriptomic signatures in your data that would support one interpretation over the other?
In the CLP population, similar features of cellular aging, characterized by telomere attrition, DNA damage and cell-cycle arrest were found (as in the HSC compartment). Our observation indicates that these cells are in a genuine senescent state. In downstream developmental trajectory, aberrant cells emerge and differentiation is dysregulated. We have deliberated on this issue in the “Discussion” section of the revised manuscript:
(Lines 355 to 364) - In the aged human CLP population, GSEA showed a significantly elevated activity in the pathways “DNA damage telomere-stress induced senescence”, “Inhibition of DNA recombination at telomere”, “DNA methylation”, and a significantly suppressed “DNA repair” process. This indicates that senescence in the early lymphoid progenitors is again characterized by DNA damage, telomere attrition, and impairment of DNA repair. In subsequent stages of the B cell development trajectory, GSEA confirms that senescence is characterized by the accumulation of cells with deficient expression of B markers, reduced DNA repair capacity, and of cells with transcriptome characteristics of memory-NK-progenitors (NK2.1 cells). Hence the mechanisms behind senescence and the respective markers at various stages of development are, with the exception of cell-cycle arrest, quite different. - Based on your findings, is it possible to propose specific markers that could enable selective elimination of this population of aging lymphoid progenitors while sparing healthy cells?
We agree that it might be possible to identify the specific differences in markers and mechanisms that could enable selectively elimination of the senescent subset. As this is the first report on the emergence of such a population in the bone marrow of aged individuals, we need further in-depth studies. We have deliberated on this issue in the “Discussion” part of the revised manuscript:
(Lines 350 to 354): Further in-depth studies on the developmental trajectory of NKPs from HSPCs to memory-like NK2.1 cells, both in human bone marrow as well as in circulating blood may provide more insight into the complex mechanisms of aging at different stages and on how to exploit this knowledge to target senescent cells in the late lymphoid trajectory. - Can your results suggest why this cluster forms specifically at this stage of differentiation? Is it a consequence of intrinsic defects within the progenitors themselves, or the result of influences from the aging niche?
Our results suggest that for the early developmental stages, senescent cells arise because of intrinsic defects such as DNA damage, telomere attrition, defective DNA repair mechanisms, etc. Further downstream in the development trajectory, we have evidence for emergence of aberrant cell clones and defects in B lymphoid differentiation that probably develop as a result of changes in the microenvironment. Obviously, this hypothesis is speculative at this juncture. Further in-depth studies are warranted, especially for the accumulation of NK progenitor-like and memory-like NKPs.
* Hennrich, M.L.; Romanov, N.; Horn, P.; Jaeger, S.; Eckstein, V.; Steeples, V.; Ye, F.; Ding, X.; Poisa-Beiro, L.; Lai, M.C.; et al. Cell-specific proteome analyses of human bone marrow reveal molecular features of age-dependent functional decline. Nat Commun 2018, 9, 4004, doi:10.1038/s41467-018-06353-4.
Round 2
Reviewer 1 Report
Comments and Suggestions for Authors
NA